# RNA sequencing revealed the multi-stage transcriptome transformations during the development of gallbladder cancer associated with chronic inflammation

Sen Yang[1☯], Litao Qin[2☯], Pan Wu[3☯], Yanbing Liu[1], Yanling Zhang[4], Bing Mao[5], Yiyang Yan[1], Shuai Yan[1], Feilong Tan[1], Xueliang Yue[1], Hongshan Liu[1], Huanzhou Xue[1]*

1 Department of Hepatobiliary and Pancreatic Surgery, People's Hospital of Zhengzhou University, Henan Provincial People's Hospital, Zhengzhou, Henan, China, 2 Medical Genetic Institute of Henan Province, Henan Provincial Key Laboratory of Genetic Diseases and Functional Genomics, People's Hospital of Zhengzhou University, Henan Provincial People's Hospital, Zhengzhou, Henan, China, 3 Department of Hepatobiliary and Pancreatic Surgery, Henan Provincial People's Hospital, School of Clinical Medicine, Henan University, Zhengzhou, Henan, China, 4 Department of Pathology, People's Hospital of Zhengzhou University, Henan Provincial People's Hospital, Zhengzhou, Henan, China, 5 Department of Clinical Research Service Center, People's Hospital of Zhengzhou University, Henan Provincial People's Hospital, Zhengzhou, Henan, China

☯ These authors contributed equally to this work.
* xhuanzhou@126.com

**Data Availability Statement:** All the datasets generated and analyzed during the current study are available from the GEO repository (accession

## Abstract

Gallbladder cancer (GBC) is a highly malignant tumor with extremely poor prognosis. Previous studies have suggested that the carcinogenesis and progression of GBC is a multi-stage and multi-step process, but most of them focused on the genome changes. And a few studies just compared the transcriptome differences between tumor tissues and adjacent noncancerous tissues. The transcriptome changes, relating to every stage of GBC evolution, have rarely been studied. We selected three cases of normal gallbladder, four cases of gallbladder with chronic inflammation induced by gallstones, five cases of early GBC, and five cases of advanced GBC, using next-generation RNA sequencing to reveal the changes in mRNAs and lncRNAs expression during the evolution of GBC. In-depth analysis of the sequencing data indicated that transcriptome changes from normal gallbladder to gallbladder with chronic inflammation were distinctly related to inflammation, lipid metabolism, and sex hormone metabolism; transcriptome changes from gallbladder with chronic inflammation to early GBC were distinctly related to immune activities and connection between cells; and the transcriptome changes from early GBC to advanced GBC were distinctly related to transmembrane transport of substances and migration of cells. Expression profiles of mRNAs and lncRNAs change significantly during the evolution of GBC, in which lipid-based metabolic abnormalities play an important promotive role, inflammation and immune activities play a key role, and membrane proteins are very highlighted molecular changes.

number, GSE202479) (https://www.ncbi.nlm.nih.gov/geo/query/acc.cgi?acc=GSE202479).

**Funding:** HL:This research was supported by a grant (SB201901079) from the Henan Province Medical Science and Technology Research Plan (http://117.160.147.213:8088). The funders had no role in study design, data collection and analysis, decision to publish, or preparation of the manuscript.

**Competing interests:** The authors have declared that no competing interests exist.

## Introduction

As a common malignant carcinoma of the biliary tree, gallbladder cancer (GBC) still has an extremely poor prognosis, with a median survival time of < 1 year [1]. The reason is that GBC has high ability to invade and metastasize, and its early diagnosis rate is quite low [2]. Currently, surgical resection is the only treatment with curative intent for GBC but very few cases are suitable for resection and most adjuvant therapy has a very low response rate, although chemotherapy, targeted therapy and immunotherapy of GBC have made great progress in recent years [3–5].

Gallbladder stone is the most important risk factor for GBC. Gallbladder stones stimulate the wall of gallbladder for a long time and cause chronic inflammation, eventually lead to carcinogenesis. This is the most commonly recognized carcinogenesis pathway in GBC [6]. Similar to other tumors, such as colon cancer, the formation and progression of GBC is a multi-stage and multi-step process with the accumulation of multiple changes in the genome and transcriptome [7–9]. As for the molecular events in this process, most previous studies have focused on genome changes, but few have involved multi-stage transcriptome changes, only a few studies just compared the transcriptome differences between tumor tissues and adjacent noncancerous tissues [10–12].

Non-coding RNAs, especially long non-coding RNAs (lncRNAs), have been a research hotspot in recent years. Except for an extremely small number of regions that encode mRNA, most parts of the human genome are still poorly understood, producing a large number of non-coding RNAs including microRNAs and lncRNAs [13, 14]. lncRNAs are a type of non-coding RNA with a length of more than 200 nucleotides. It can perform physiological functions through various mechanisms, such as trans- and cis-regulation. An increasing number of studies has shown that lncRNAs are associated with a variety of diseases, including cancer [15–17]. However, little is known about its role in carcinogenesis and progression of GBC [18].

The improvement of the treatment effect of GBC depends on the development of more effective drugs, which requires further understanding of the molecular mechanism of carcinogenesis and progression of GBC. Therefore, using next-generation sequencing technology, we studied the expression profiles of mRNA and lncRNAs in the four stages of this process: normal gallbladder, gallbladder with chronic inflammation, early GBC, and advanced GBC. Through cluster analysis, we identified the highlighted molecular category changes during GBC development.

## Materials and methods

### Case selection and sample processing

A number of gallbladder and GBC samples were collected from Henan Provincial People's Hospital between 2019-08-16 and 2020-12-31. Our study was approved by the Ethics Committee of Henan Provincial People's Hospital and started on 2019-08-16. Written informed consent was obtained from all the participants. And their capacity to provide consent was assessed by the researchers. All the participants were adults with age range between 40 and 82. They were in well performance status, spoke and understood Chinese, and were able to give informed consent. This consent procedure was approved by the ethics committee. Additionally, all methods were performed in accordance with the relevant guidelines and regulations. We had access to information that could identify individual participants during or after data collection. Specimens were immediately frozen in liquid nitrogen after resection and stored at -80°C for long-term storage. Considering the requirements of scientific experiments on repeatability, the difficulty of obtaining some types of samples, and the high requirements of

RNA sequencing experiments on sample quality, eventually we confirmed three cases of normal gallbladder (N8 N10 N20), four cases of gallbladder with chronic inflammation (Y8 Y12 Y13 Y16), five cases of GBC in the early stage (T5 T12 T13 T18 T31), and five cases of GBC in the advanced stage (T1 T19 T22 T27 T32), and performed transcriptome sequencing (Shanghai Biotechnology Company, Shanghai, China). The sample selection principles were as follows: normal gallbladder specimens were obtained from patients who underwent hepatectomy or pancreaticoduodenectomy without stones, polyps, obstructive jaundice, or cholangitis; the chronic inflamed gallbladder specimens were surgically removed from patients with calculous cholecystitis, excluding acute cholecystitis; GBC specimens should be adenocarcinoma pathologically. All specimens were pathologically confirmed. GBC samples were staged according to the AJCC 8th edition TNM staging method. Stages 0, , and  were defined as early stage, and stages III–IV were considered as advanced stage. Clinicopathological data of the selected samples are shown in S1 and S2 Tables.

## RNA extraction and quality inspection

The TransZol Up Plus RNA Kit (Cat#ER501-01, Trans, Beijing, China) was used for total RNA extraction according to the manufacturer's instructions. Total RNA was purified using an RNAClean XP Kit (Cat A63987, Beckman Coulter, Inc. Kraemer Boulevard Brea, CA, USA) and RNase-Free DNase Set (Cat#79254, QIAGEN, GmBH, Germany) after passing quality inspection using an Agilent Bioanalyzer 2100 (Agilent Technologies, Santa Clara, CA, US). Purified total RNA was subjected to quality inspection using a NanoDrop ND-2000 spectrophotometer and Agilent Bioanalyzer 2100 (Agilent Technologies, Santa Clara, CA, US). Finally, only qualified total RNA was used for subsequent sequencing experiments.

## Sequencing experiments

First, the purified total RNA was subjected to rRNA removal, fragmentation, first-strand cDNA synthesis, second-strand cDNA synthesis, end repair, 3'end addition, connector ligation, and enrichment to build a sequencing sample library following the experimental instructions. The concentration of the constructed library was detected using a Qubit® 2.0 Fluorometer, and the size of the library was detected using the Agilent 2100. The reagents used for library construction and quality inspection are listed in S3 Table. The quality inspection results are presented in S4 Table.

Then, cluster generation and first-direction sequencing primer hybridization were performed on the cBot equipped with the Illumina sequencer (Illumina NovaSeq 6000), following the cBot User Guide.

Finally, the flow cell with the cluster was placed on the sequencing machine using the prepared sequencing reagents, according to the Illumina User Guide. A paired-end program was used to perform paired-end sequencing. The sequencing process was controlled by data collection software provided by Illumina, and real-time data analysis was performed. The quality control standard for sequencing results was as follows: the amount of data was about 10G/sample, and the ratio of base quality in each direction greater than 20 (Q20) was not less than 85%. Sequencing quality was evaluated by the Q value, and the relationship between the Q value and sequencing error rate E value is

$$Q = -10Log_{10}E$$

The sequencing quality of all samples was excellent and the base distribution was balanced. The quality control results are presented in S5 Table.

## RNA sequencing data analysis

The raw reads obtained by sequencing may contain unqualified reads with low end quality and sequencing primers. These unqualified reads may have a certain impact on the quality of the analysis; therefore, they must be filtered to obtain clean reads for data analysis. We used Seqtk (https://github.com/lh3/seqtk) to filter raw reads, according to the following procedure: 1. removal of the ligation sequence; 2. removal of the bases whose 3'end quality Q is less than 20; 3. removal of reads with a length of less than 25 bp; 4. removal of the ribosome RNA reads from each species. The pre-processed statistics are presented in S6 Table.

Genome mapping was performed on the pre-processed reads using a spliced mapping algorithm from Hisat2 (version:2.0.4) [19]. The genome version used was GRCh38. The mapping process adopted the default parameters. The mapping results are listed in S7 Table.

To make the gene expression levels of different genes and samples comparable, the reads were converted into FPKM (fragments per kilobase of exon model per million mapped reads) to standardize gene expression [20]. We first used Stringtie (version: 1.3.0) [21, 22] to count the number of fragments of each gene after Hisat2 alignment, then used the trimmed mean of M values (TMM) method to normalize them [23], and finally calculated the FPKM value of each gene through a Perl script. The FPKM formula is as follows:

$$\text{FPKM} = \frac{\text{total exon fragments}}{\text{mapped reads(millions)} \ \times \text{exon length(KB)}}$$

Total exon fragments are the number of fragments aligned to the gene exon (fragment: a pair of reads); exon length is the total length of the gene exon; and mapped reads are the total number of reads aligned to the reference genome.

Differentially expressed genes were analyzed using edgeR [24]. The obtained p-value was subjected to multiple hypothesis tests, and the adjusted p-value was called the q-value. The p-value threshold was determined by controlling the false discovery rate. Furthermore, we calculated the multiple of differential expression based on the FPKM value, namely fold-change. The screening conditions for the differentially expressed genes were as follows: 1. q-value $\leq$ 0.05; 2. fold-change $\geq$ 2.

## Function analysis for differentially expressed genes

Using GO (Gene Ontology) (http://www.geneontology.org/) analysis, the number of differentially expressed genes with the same function term was calculated. KEGG (Kyoto Encyclopedia of Genes and Genomes) (http://www.kegg.jp/) analysis was used to count the number of differentially expressed genes in each pathway. Furthermore, GO and KEGG enrichment analyses were performed to screen for significantly enriched GO and KEGG terms from the differentially expressed genes. The calculation formula for the p-value is as follows:

$$P = 1 - \sum_{i=0}^{m-1} \frac{\binom{M}{i}\binom{N-M}{n-i}}{\binom{N}{n}}$$

The calculation formula for rich factor was as follows: rich factor = (m/ n)/ (M/ N).

N is the number of genes with GO or KEGG annotation among all genes, n is the number of differentially expressed genes in N, M is the number of genes annotated as a specific GO or KEGG term among all genes, and m is the number of differentially expressed genes annotated

as a specific GO or KEGG term. The q-value was obtained from the p-value after the multiple hypothesis test. With q-value ≤ 0.05, the GO or KEGG terms that satisfied this condition were defined as significantly enriched in differentially expressed genes. The smaller the q-value, the more significant the enrichment. The greater the rich factor, the greater the degree of enrichment.

## LncRNA analysis

The spliced results of Stringtie (version 1.3.0) were compared with the reference annotations using gffcompare (version 0.9.8), and new transcripts that failed to match the known annotations were obtained. Three types of transcripts (i.e., i, u, and x) were extracted for lncRNA prediction. The specific steps were as follows: step1: transcription length ≥ 200bp and exon ≥ 2; step2: predicted ORF < 300bp; step3: predict using Pfam [25], CPC [26], CNCI [27], and select the transcripts with CPC score <0 and CNCI score <0 and insignificant Pfam comparison as the potential lncRNAs; and step 4: compare with known lncRNAs and remove the same sequence. Remarks: i: a transfrag falling entirely within a reference intron; u: unknown, intergenic transcript; x: exonic overlap with reference on the opposite strand.

Expression quantification was performed for the predicted novel and known lncRNAs from the NONCODE and Ensembl database. The ID starting with MSTRG is a novel lncRNA, the ID starting with NON is the known lncRNA in the NONCODE database, and the ID starting with ENS is the known lncRNA in the Ensembl database.

Trans- and cis-regulation was used to predict target genes. The mRNA database of this species was used for trans-prediction. First, BLAST was to select complementary or similar sequences, then RNAplex [28] was used to calculate the complementary energy between the two sequences, and finally, sequences above the threshold were selected. Genes whose distance from the lncRNA was less than 10 kb were selected as the target genes for cis regulation.

## Quantitative real-time PCR

To further verify the accuracy of the RNA sequencing experiment, quantitative real-time PCR was performed. Two differentially expressed mRNAs and two differentially expressed lncRNAs were selected for each comparison, and a total of 12 genes were determined for this test. The specimens used were the same as those used in the sequencing experiment.

RNA was extracted as described above. Quantitative real-time experiments were performed using Power SYBR Green PCR Master Mix (Cat#4368708, ABI, USA) according to the manufacturer's instructions. The primer sequences of the related genes are listed in S8 Table. β-Actin was used as the reference gene. Each reaction was performed in triplicates. Relative expression of each gene was quantified using the gene's $2^{-\Delta Ct}$.

## Statistical analysis

All statistical analyses were performed using SPSS for Windows, version 24.0. The analytical methods used in the sequencing experiments were described above. Quantitative real-time PCR results were compared between groups using an independent sample t-test. The expression levels of the genes in each group are shown as mean ± standard deviation. Statistical significance was set at P ≤ 0.05.

## Results

### Overview of sequencing results and verification by quantitative real-time PCR

Compared with the human genome, the ratio of reads aligned to gene regions, coding regions, splice sites, introns, and non-coding regions was normal, genome coverage was good, and sequencing quantity was sufficient, as shown in S1 Fig.

A total of 84043 lncRNAs were detected, of which 1030 lncRNAs were newly predicted. By observing the differences in transcript length, number of exons, and expression levels between lncRNAs and mRNAs, it was shown that the lncRNAs conformed to the general characteristics, as shown in S2 Fig.

Considering the uniformity and quantity of samples, we selected 12 genes with significant expression differences between groups for qPCR verification experiment, including CYP1A1 IGF1 ENST00000555772 NONHSAT247740.1 for comparison normal gallbladder VS inflammatory gallbladder, C4BPB PRKCB ENST00000648838 NONHSAT104346.2 for comparison inflammatory gallbladder VS early GBC, and HLA-DRB5 SLC7A5 NONHSAT159810.1 NONHSAT225391.1 for comparison early GBC VS advanced GBC. This experiment indicated that

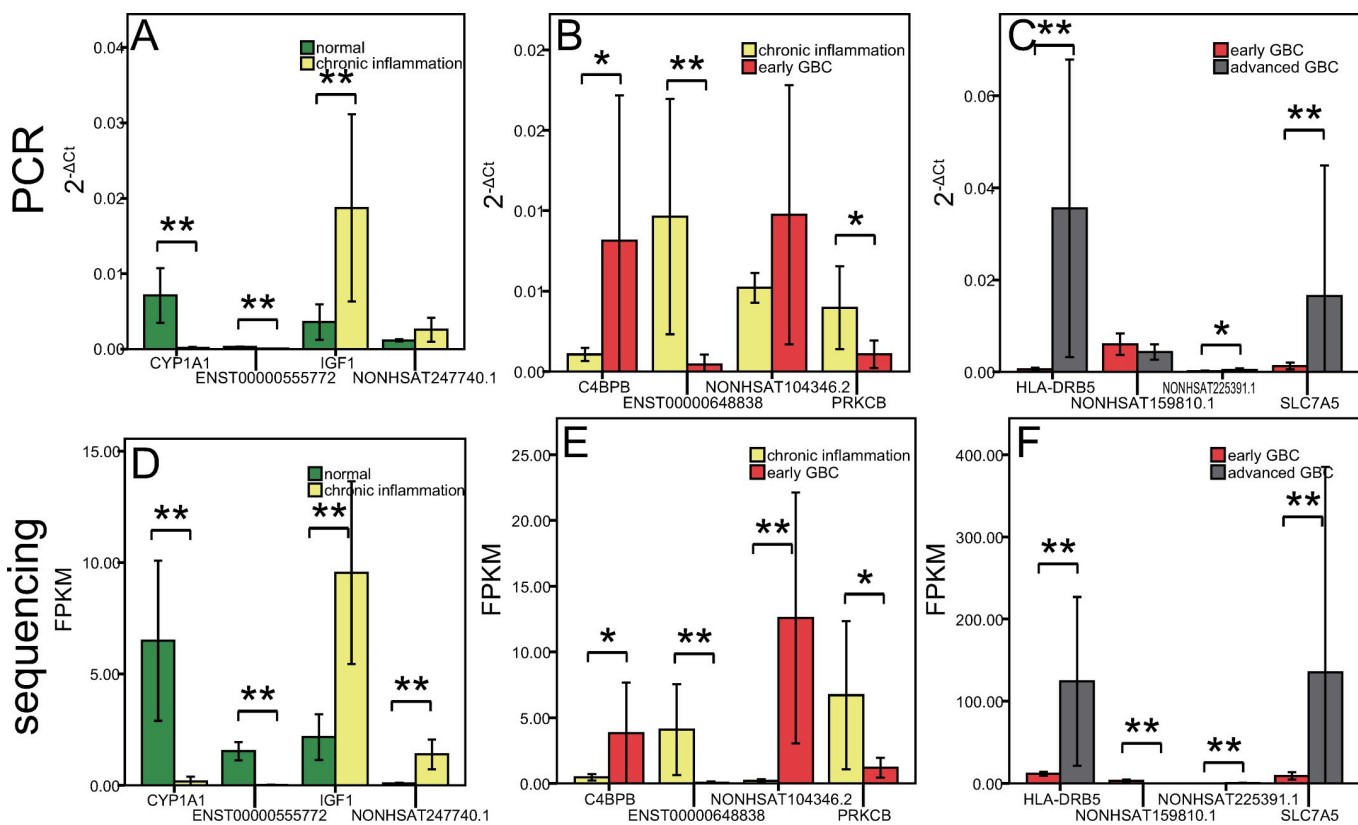

**Fig 1. Verification through quantitative real-time PCR experiment.** The first row of three graphs are the results of the qPCR experiment. The X axis represents each gene, and the Y axis represents the relative expression of each gene in the PCR experiment, represented as $2^{-\Delta Ct}$ mean ± standard deviation. The second row of three graphs are the results of the RNA sequencing experiments. The X axis represents each gene, and the Y axis represents the relative expression of each gene in the sequencing experiment, represented as FPKM mean ± standard deviation. (A) and (D) are the results of the four genes selected in the comparison between normal gallbladder and chronic inflammation gallbladder. (B) and (E) are the results of the four genes selected in the comparison between chronic inflammation gallbladder and early GBC. (C) and (F) are the results of the four genes selected in the comparison between early GBC and advanced GBC. * $P \leq 0.05$ ** $P \leq 0.01$.

the qPCR results were highly consistent with the sequencing results, suggesting that the sequencing experiment had high reliability. As shown in Fig 1.

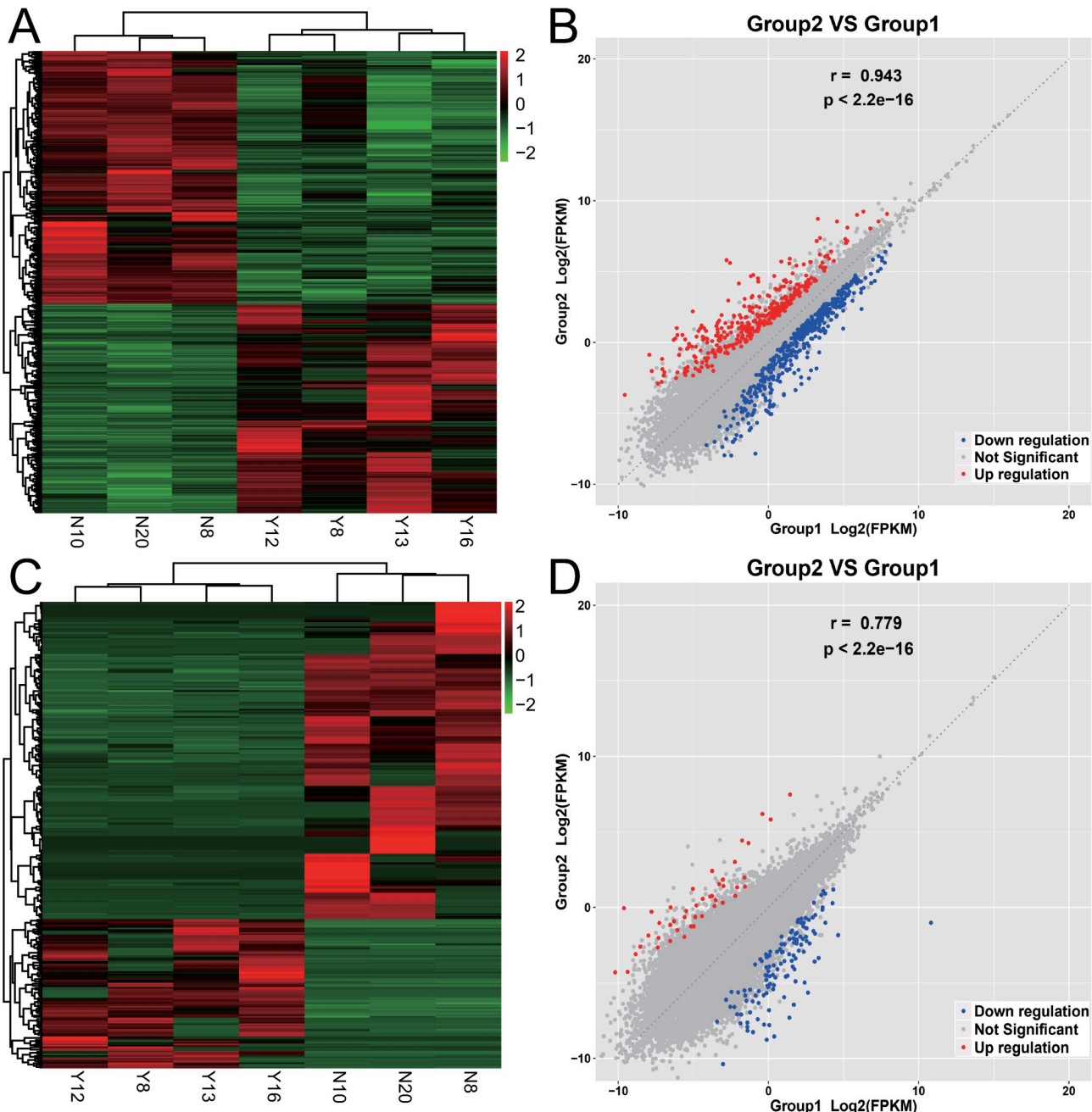

**Fig 2. Expression differences of mRNAs and lncRNAs between normal gallbladders and gallbladders with chronic inflammation.** Group 1 is normal gallbladder including N8 N10 N20; Group 2 is gallbladder with chronic inflammation including Y8 Y12 Y13 Y16. (A) The heatmap figure of mRNA expression between the two groups. The deeper the red, the higher the expression, and the darker the green, the lower the expression. (B) The correlation scatter diagram of mRNA expression between the two groups, the red dots are the upregulated mRNAs of gallbladder with chronic inflammation relative to the normal gallbladder, the blue dots are the downregulated mRNAs, and the gray dots indicate the differences are not significant. (C) The heatmap figure of lncRNA expression between the two groups. (D) The correlation scatter diagram of lncRNA expression between the two groups.

## Transcriptome changes from normal gallbladder to gallbladder with chronic inflammation

A total of 851 different mRNAs were identified, of which 385 were upregulated and 466 were downregulated. There were 322 different lncRNAs, of which 103 were upregulated and 219 were downregulated. The expression of mRNAs and lncRNAs showed obvious differences between the two groups, and the expression of samples in the same group showed good homogeneity, as shown in Fig 2.

**GO enrichment of differentially expressed mRNAs.** GO enrichment analysis revealed 759 GO terms with q-value ≤ 0.05, which indicated significant gene functions of differentially expressed genes preliminarily. Because similar GO terms can be categorized into larger categories, the top 100 GO terms with a larger rich factor were further classified in order to discover the highlighted gene function categories of differentially expressed genes. It was found that differentially expressed mRNAs were distinctly related to inflammation (35 terms), metabolism (14 terms) including lipid metabolism (10 terms) and sex hormone metabolism (four terms). The top three enriched GO terms were estrogen 16-α-hydroxylase activity, lipid hydroxylation, and the omega-hydroxylase P450 pathway, as shown in Fig 3.

**KEGG enrichment of differentially expressed mRNAs.** KEGG enrichment analysis revealed 28 KEGG terms with q-value ≤ 0.05, which indicated significant pathways that differentially expressed genes took part in. Through further classifying this 28 KEGG terms, it was found that differentially expressed mRNAs were distinctly related to inflammation (six terms), lipid metabolism (three terms), steroid hormones metabolism (three terms), amino acid and foreign substance metabolism (seven terms). This was similar to the GO enrichment result. They both indicated that the transcriptome differences between normal gallbladder and gallbladder with chronic inflammation were distinctly related to inflammation and metabolism. The top three enriched KEGG terms were phenylalanine, tyrosine and tryptophan biosynthesis, synthesis and degradation of ketone bodies, and steroid hormone biosynthesis, as shown in Fig 3.

**GO and KEGG enrichment of differentially expressed lncRNAs.** Target genes were predicted by trans- and cis-regulation. There were 877 predicted target genes for the differentially expressed lncRNAs, of which 59 showed significant differences in expression.

There were 0 GO terms with q-value ≤ 0.05 for target genes, and there were two KEGG terms with q-value ≤ 0.05, which were homologous recombination, valine, leucine, and isoleucine degradation.

GO and KEGG enrichment analyses was also performed for differentially expressed target genes. There were 28 GO terms with q-value ≤ 0.05, which were distinctly related to inflammation (17 terms) and foreign substance metabolism (four terms). There were seven KEGG terms with q-value ≤ 0.05, and 16 terms with p-value ≤ 0.05, which were mostly related to inflammation (nine terms), lipid metabolism (two terms), and tumor-related pathways (three terms), as shown in S3 Fig.

## Transcriptome changes from gallbladder with chronic inflammation to early GBC

A total of 176 different mRNAs were identified, of which 58 were upregulated and 118 were downregulated. There were 84 different lncRNAs that were identified, of which 20 were upregulated and 60 were downregulated. The expression of mRNAs and lncRNAs showed obvious differences between the two groups, and the expression of samples in the same group showed good homogeneity, as shown in Fig 4.

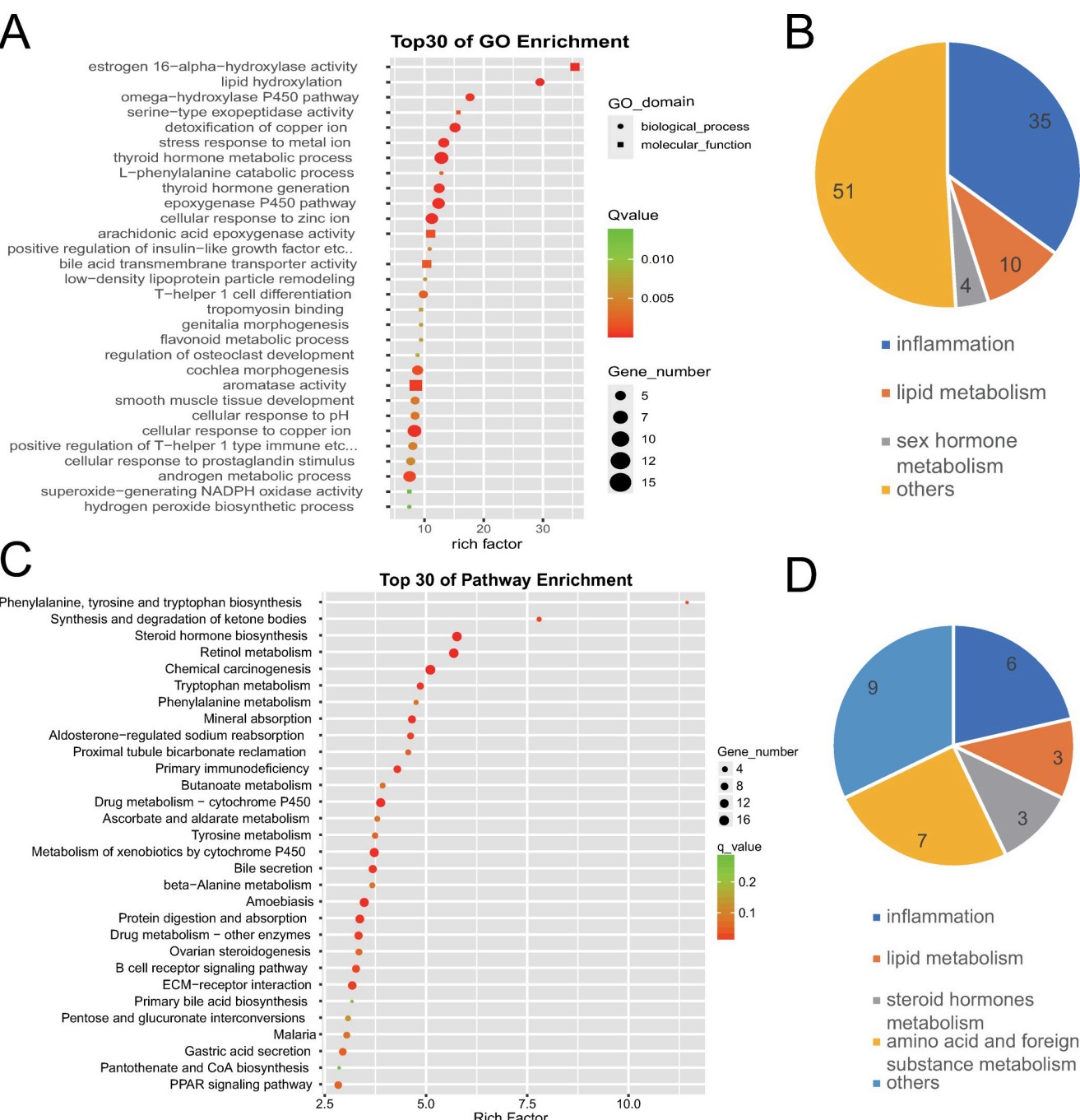

**Fig 3. GO and KEGG enrichment analysis of differentially expressed mRNAs between normal gallbladders and gallbladders with chronic inflammation.** (A) The top 30 GO terms with a high degree of enrichment. The shapes of icons represent different GO categories, the size of icons represents the number of differentially expressed genes contained by this GO term, the color depth represents the size of q-valueand the X axis indicates the value of the rich factor. (B) The top 100 GO terms with a larger enrichment factor were further classified. Numbers on the graph represent the number of GO terms corresponding to the category. (C) The top 30 KEGG terms with a high degree of enrichment. The size of icons represents the number of differentially expressed genes contained by this KEGG term, the color depth represents the size of q-value, and the X axis indicates the value of the rich factor. (D) The 28 KEGG terms with q-value ≤ 0.05 were further classified. Numbers on the graph represent the number of KEGG terms corresponding to the category.

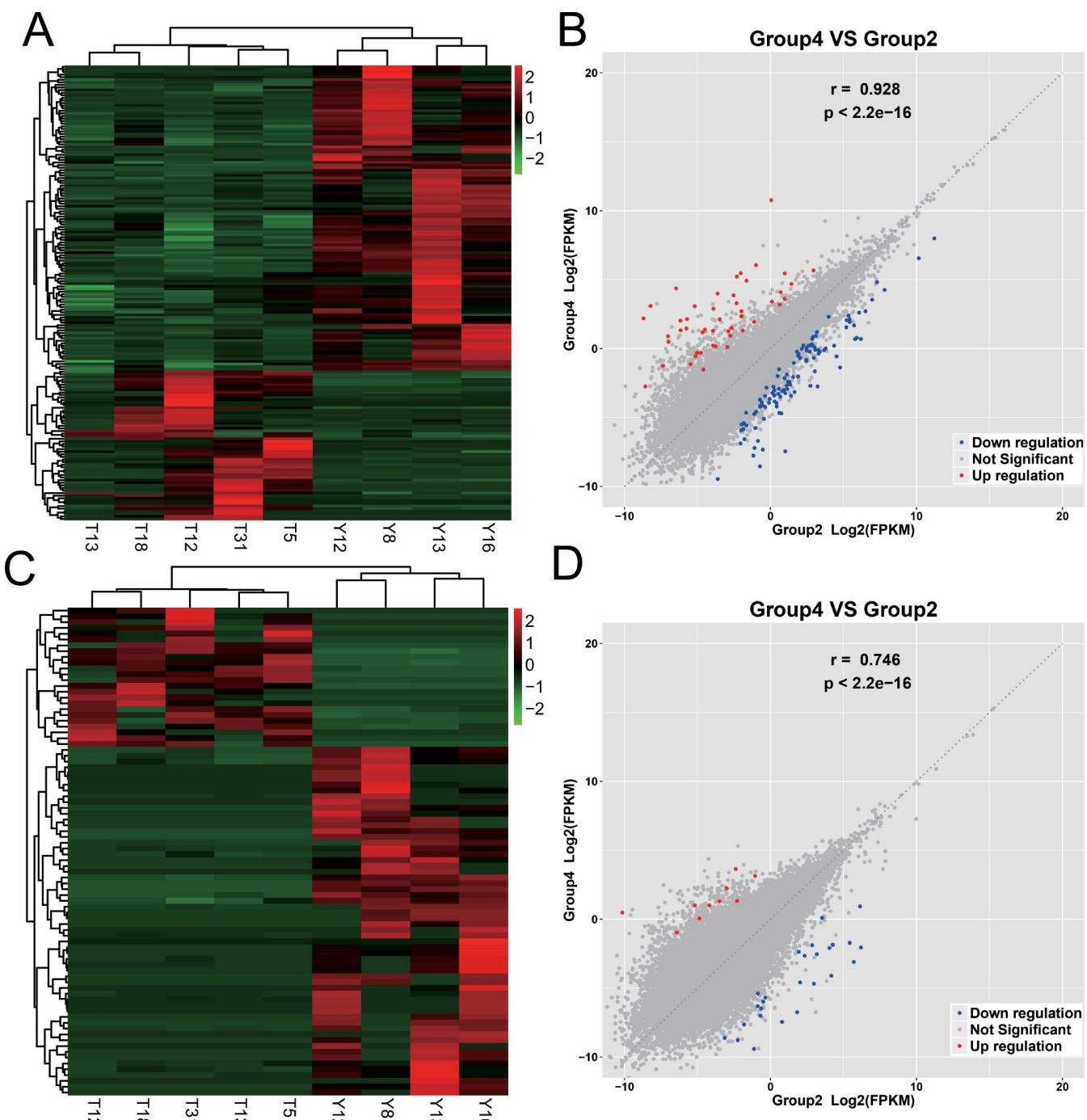

**Fig 4. Expression differences of mRNAs and lncRNAs between gallbladders with chronic inflammation and early GBC.** Group 2 is gallbladder with chronic inflammation including Y8 Y12 Y13 Y16; group 4 is early GBC including T5 T12 T13 T18 T31. (A) The heatmap figure of mRNA expression between the two groups. The deeper the red, the higher the expression, and the darker the green, the lower the expression. (B) The correlation scatter diagram of mRNA expression between the two groups, the red dots are the upregulated mRNAs of gallbladder with early GBC relative to gallbladder with chronic inflammation, the blue dots are the downregulated mRNAs, and the gray dots indicate the differences are not significant. (C) The heatmap figure of lncRNA expression between the two groups. (D) The correlation scatter diagram of lncRNA expression between the two groups.

**GO enrichment of differentially expressed mRNAs.** GO enrichment analysis revealed 116 GO terms with q-value ≤ 0.05, which indicated significant gene functions of differentially

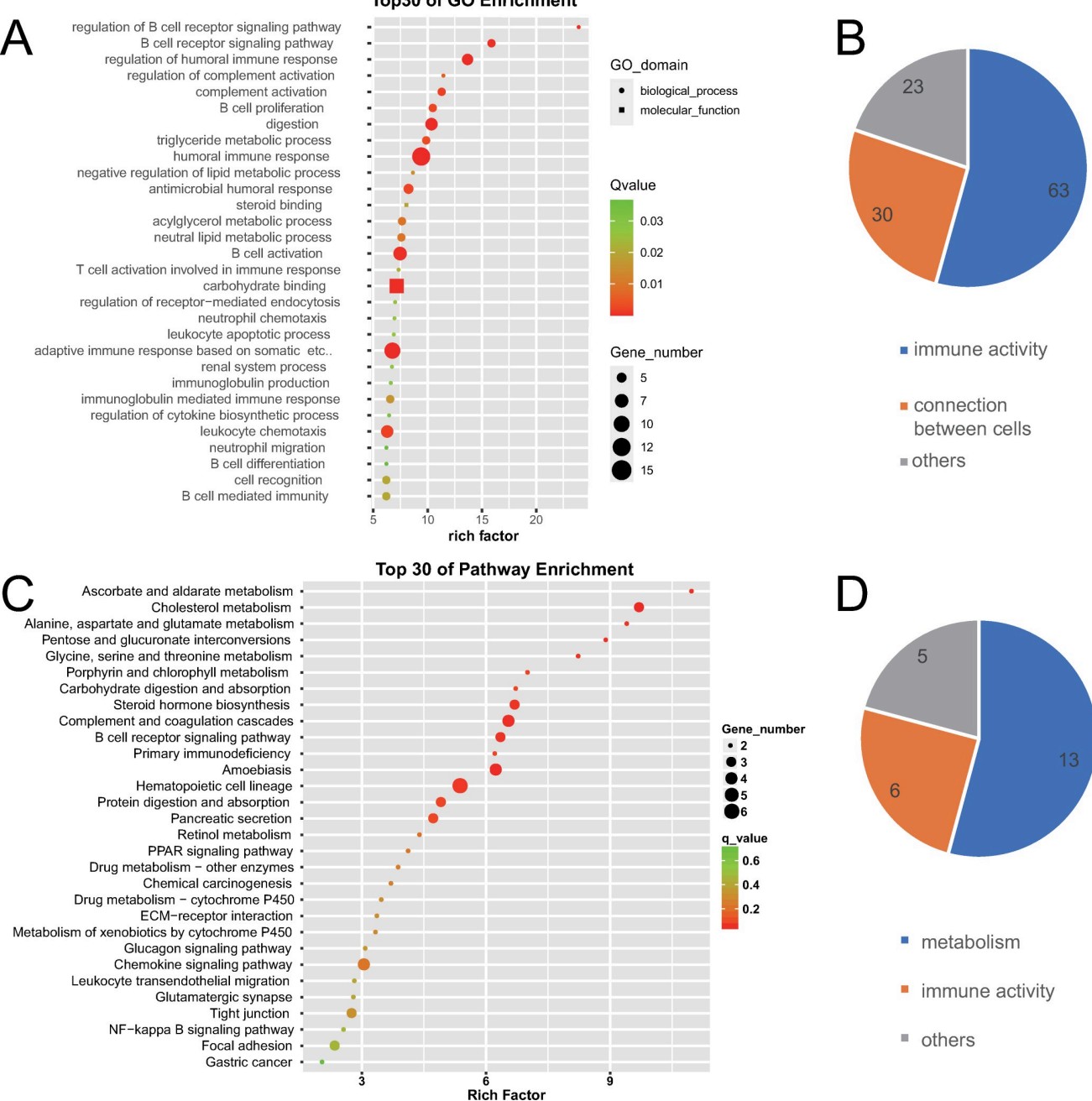

**Fig 5. GO and KEGG enrichment analysis of differentially expressed mRNAs between gallbladders with chronic inflammation and early GBC.** (A) The top 30 GO terms with a high degree of enrichment. The shapes of icons represent different GO categories, the size represents the number of differentially expressed genes contained by this GO term, the color depth represents the size of the q-valueand the X axis indicates the value of the rich factor. (B) The 116 GO terms with q-value ≤ 0.05 were further classified. Numbers on the graph represent the number of GO terms corresponding to the category. (C) The top 30 KEGG terms with a high degree of enrichment. The size of icons represents the number of differentially expressed genes contained by this KEGG term, the color depth represents the size of q-value, and the X axis indicates the value of the rich factor. (D) The 24 KEGG terms with p-value ≤ 0.05 further classified. Numbers on the graph represent the number of KEGG terms corresponding to the category.

expressed genes preliminarily. Further cluster analysis revealed that this 116 GO terms were distinctly related to immune activity (63 terms) and connection between cells (30 terms), which further indicate the highlighted gene function categories of differentially expressed genes. The top three enriched terms were regulation of B cell receptor signaling pathway,

regulation of humoral immune response, and regulation of complement activation, as shown in Fig 5.

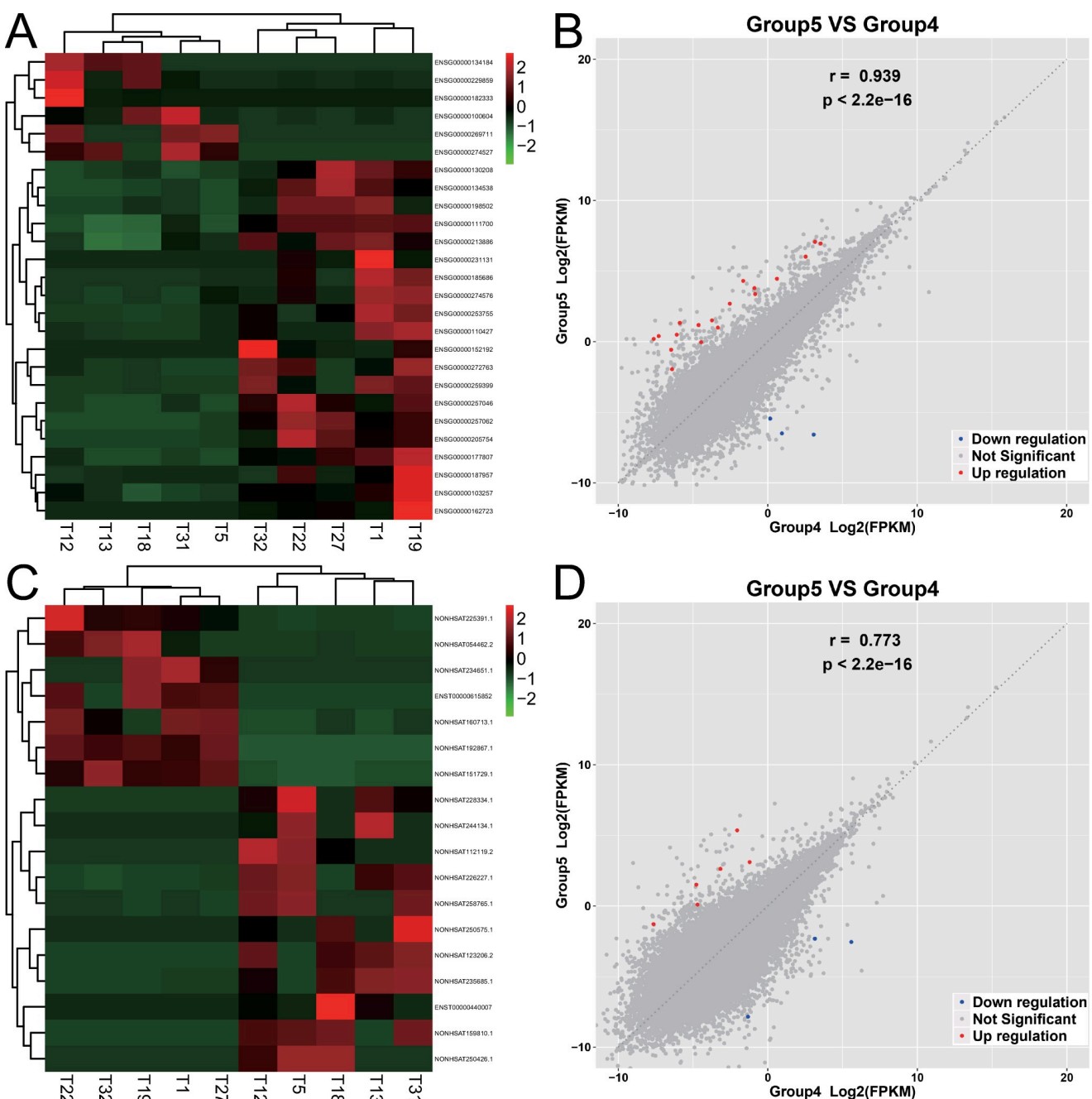

**Fig 6. Expression differences of mRNAs and lncRNAs between early GBC and advanced GBC.** Group 4 is early GBC including T5 T12 T13 T18 T31. Group 5 is advanced GBC including T1 T19 T22 T27 T32. (A) The heatmap figure of mRNA expression between the two groups. The deeper the red, the higher the expression, and the darker the green, the lower the expression. (B) The correlation scatter diagram of mRNA expression between the two groups, the red dots are the upregulated mRNAs of advanced GBC relative to early GBC, the blue dots are the downregulated mRNAs, and the gray dots indicate the differences are not significant. (C) The heatmap figure of lncRNA expression between the two groups. (D) The correlation scatter diagram of lncRNA expression between the two groups.

**KEGG enrichment of differentially expressed mRNAs.**   There were seven KEGG terms with q-value $\leq 0.05$, and 24 terms with p-value $\leq 0.05$, which indicated significant pathways that differentially expressed genes took part in. Further cluster analysis revealed that this 24 terms were mostly related to metabolism (13 terms) and immune activity (six terms), as shown in Fig 5. This was partially similar to the GO enrichment result which also indicated immune activity was significant transcriptomic difference between gallbladder with chronic inflammation and early GBC.

**GO and KEGG enrichment of differentially expressed lncRNAs.**   Target genes were predicted by trans- and cis-regulation. There were 54 predicted target genes for differentially expressed lncRNAs, of which six showed significant differences in expression.

There were 0 GO terms with q-value $\leq 0.05$ and 76 terms with p-value $\leq 0.05$ for target genes, which were mostly related to the modification and polymerization of proteins (36 terms), connection and signal transduction between cells (23 terms), and immune activity (seven terms). There were 0 KEGG terms with q-value $\leq 0.05$, and 17 terms with p-value $\leq 0.05$, which were mostly related to immune activity (eight terms) and signal transduction (six terms), as shown in S4 Fig.

GO and KEGG enrichment analyses were also performed for the differentially expressed target genes. There were 0 GO terms with q-value $\leq 0.05$, and five terms with p-value $\leq 0.05$, which were related to development (four terms) and connection between cells (one terms). There was 0 KEGG terms with q-value $\leq 0.05$, and 0 terms with p-value $\leq 0.05$.

## Transcriptome changes from early GBC to advanced GBC

A total of 26 different mRNAs were identified, of which 20 were upregulated and six were downregulated. There were 18 different lncRNAs, of which seven were upregulated and 11 were downregulated. The expression of mRNAs and lncRNAs showed obvious differences between the two groups, and the expression of samples in the same group showed good homogeneity, as shown in Fig 6.

**GO enrichment of differentially expressed mRNAs.**   GO enrichment analysis revealed 11 GO terms with q-value $\leq 0.05$, which indicated significant gene functions of differentially expressed genes preliminarily. Significantly, further cluster analysis revealed that this 11 GO terms were all related to the transmembrane transport of substances (11 terms), including the transmembrane transport of carboxylic acids (three terms), ions (six terms), and phospholipids (two terms), which further indicate the highlighted gene function categories of differentially expressed genes. Cluster analysis was further expanded to the 25 terms with p-value $\leq 0.05$, which were distinctly related to transmembrane transport of substances (17 terms), cell membrane components (three terms), and cell migration (three terms), which was similar to the cluster analysis of the 11 GO terms with q-value $\leq 0.05$. The top three enriched terms were carboxylic acid transmembrane transporter activity, carboxylic acid transmembrane transport, and organic anion transmembrane transporter activity, as shown in Fig 7.

**KEGG enrichment of differentially expressed mRNAs.**   KEGG enrichment analysis revealed only 1 KEGG term with q-value $\leq 0.05$ or p-value $\leq 0.05$, which indicated the most significant pathway that differentially expressed genes took part in. This significant KEGG term was bile secretion, as shown in Fig 7. This was partially similar to the GO enrichment result, because that bile secretion activity involves transmembrane transport of multiple substances.

**GO and KEGG enrichment of differentially expressed lncRNAs.**   Target genes were predicted by trans- and cis-regulation. There were 14 predicted target genes for the differentially expressed lncRNAs, none of which showed significant differences in expression.

There were three GO terms with q-value $\leq 0.05$ and 50 terms with p-value $\leq 0.05$ for target genes, which were mostly related to RNA expression regulation (19 terms), cell proliferation (five terms), and cell migration (three terms). There was only one KEGG term with q-value $\leq 0.05$ or p-value $\leq 0.05$, that was miRNAs in cancer, as shown in S5 Fig.

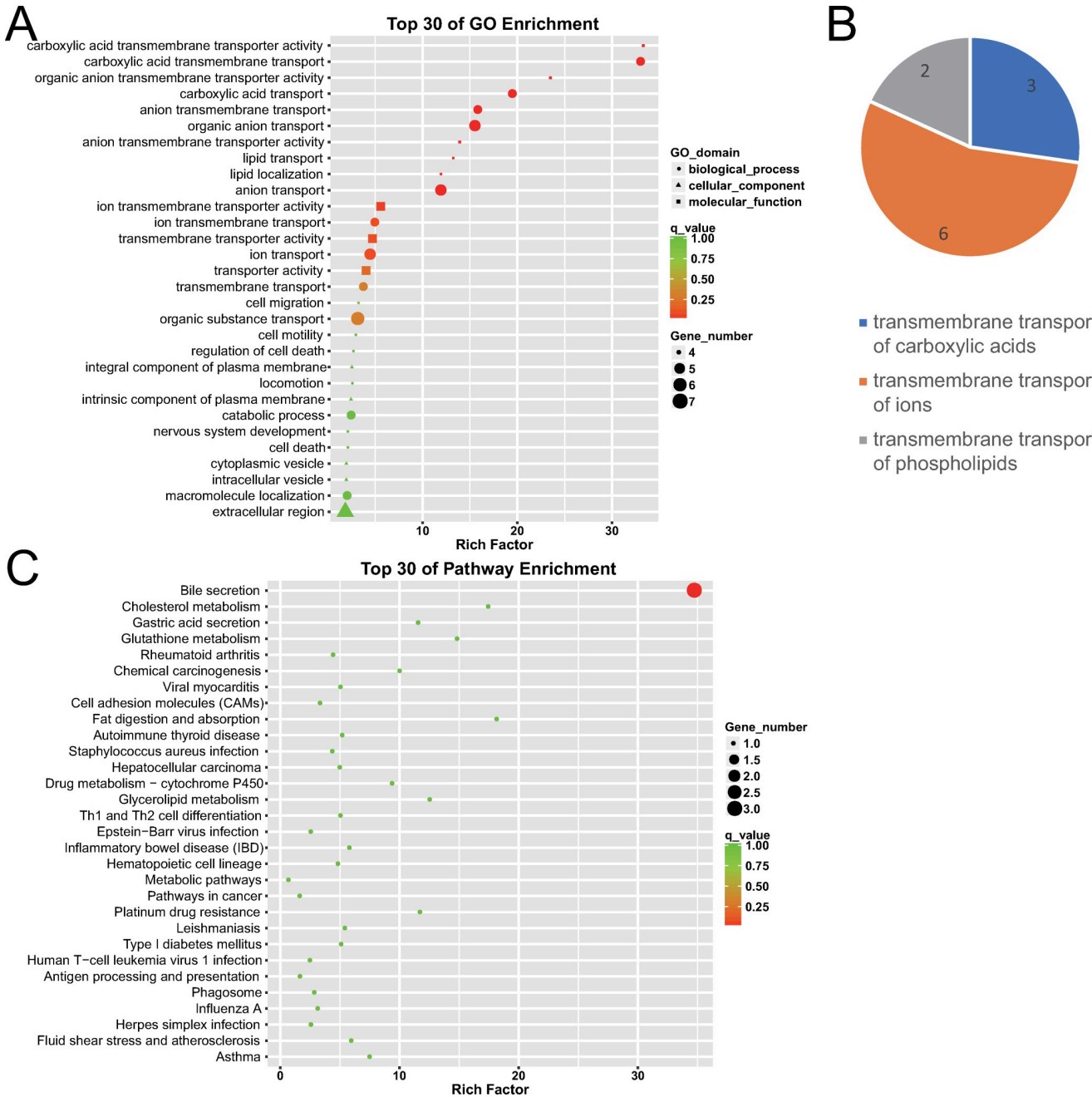

**Fig 7. GO and KEGG enrichment analysis of differentially expressed mRNAs between early GBC and advanced GBC.** (A) The top 30 GO terms with a high degree of enrichment. The shapes of icons represent different GO categories, the size represents the number of differentially expressed genes contained by this GO term, the color depth represents the size of the q-valueand the X axis indicates the value of rich factor. (B) The 11 GO terms with q-value $\leq 0.05$ were further classified. Numbers on the graphrepresent the number of GO terms corresponding to the category. (C) The top 30 KEGG terms with a high degree of enrichment. The size of icons represents the number of differentially expressed genes contained by this KEGG term, the color depth represents the size of q-valueand the X axis indicates the value of the rich factor.

## Discussion

Previous studies have suggested that the carcinogenesis and progression of GBC is a multi-stage and multi-step process, but most of them focused on the genome level. The transcriptome level, such as changes in the expression profiles of mRNAs and lncRNAs, have rarely been studied. To this end, we selected normal human gallbladder, chronically inflamed gallbladder, early GBC, and advanced GBC tissue samples; performed transcriptome sequencing of mRNAs and lncRNAs; and explored the expression profile transformations of GBC during the evolution of GBC. For differentially expressed genes, we performed GO and KEGG enrichment analyses. Generally, the adjusted q-value $\leq 0.05$ was used as the significance threshold. If there were fewer corresponding terms, then p-value $\leq 0.05$ was used as the significance threshold instead, although it was not stricter than the q-value standard. For the significant terms, we further classified them one by one to discover the underlying highlighted gene functions behind the differentially expressed genes.

We found that GO and KEGG enrichment analysis had similar results. Comprehensive analysis of GO and KEGG enrichment results showed that the transcriptome differences between normal gallbladder and gallbladder with chronic inflammation were distinctly related to inflammation, lipid metabolism, and sex hormone metabolism; the transcriptome differences between gallbladder with chronic inflammation and early GBC were distinctly related to immune activities and connection between cells; the transcriptome differences between early and advanced GBC were distinctly related to transmembrane transport of substances and migration of cells.

Our study revealed that lncRNA transcription changed significantly during the formation and evolution of GBC, which was consistent with previous studies [29, 30]. Trans and cis regulation are important regulatory methods for lncRNAs [31]. We used this mechanism to predict the target genes of differentially expressed lncRNAs and then performed GO and KEGG analyses. These results were consistent with the mRNA analysis results in some aspects, but there were also differences. The reason may be that current research on lncRNAs is still in its infancy, and the functions of most lncRNAs are still unclear. Thus, it was not possible to perform functional analysis of differentially expressed lncRNAs directly; we could only use lncRNA target genes for indirect analysis, which was not rigorous in fact. In addition, lncRNAs should have many other targeted genes through other regulation mechanisms.

It is believed that gallbladder stone is the most important cause of GBC, and obesity, metabolic syndrome, and sex are also important risk factors for GBC [6, 9]. We found that the transcriptome differences between the inflammatory gallbladder and the normal gallbladder were distinctly related to inflammation, lipid metabolism, and sex hormone metabolism, which was consistent with previous studies. However, it was not clear whether the changes in metabolism-related genes that were mainly related to lipid metabolism were secondary changes after the formation of stones or whether such populations had changes in these metabolic genes, which led to the formation of stones. After all, obesity and metabolic syndrome are also risk factors for gallstone formation [9]. In addition, the metabolism of sex and other steroid hormones is a type of lipid metabolism. It is not clear whether the changes in metabolism-related genes contributed to changes in the levels of sex and other steroid hormones, or whether changes in sex and other steroids hormones affected the lipid-based metabolic changes. It may be a causal relationship, or it may be a kind of synergistic effect.

We found that changes related to inflammation were important in gallbladders with chronic inflammation, and changes related to immune activity were important in early GBC. Inflammation and immune activity are closely related and share many similarities in many ways. Therefore, it can be said that inflammation and immune activity play a key role in the

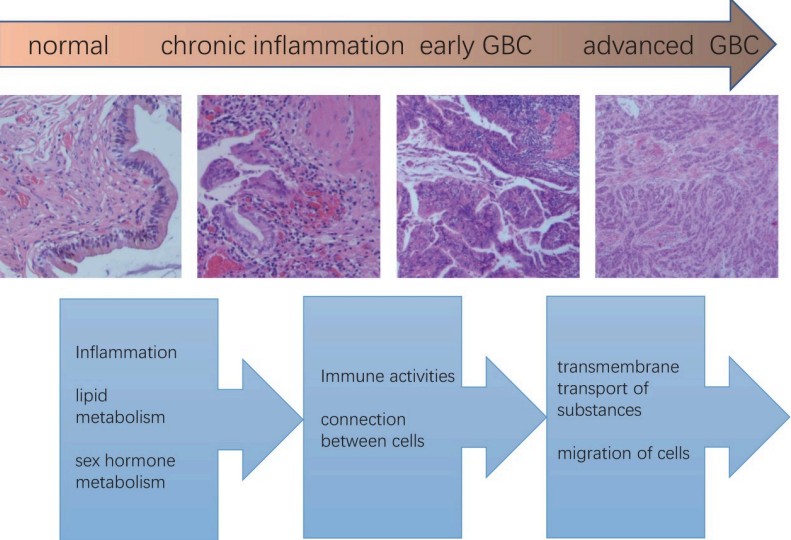

**Fig 8. Schematic diagram of the multi-stage development of GBC.** The top arrow indicates the four stages in the development of GBC. The four pictures in the middle were taken from typical pictures of each stage from our specimens. The text below indicates the prominent gene expression changes during the phase transition.

formation of GBC. In addition, it is widely known that connection (or communication) between cells is an extremely important molecular activity in the process of inflammatory and immune activity and membrane proteins are the most important participants in communication between cells. Membrane proteins also play extremely important role in the transportation of substances across the membrane and migration of cells. In short, membrane proteins play an important role in all these activities including inflammation, immune activity, connection between cells, transmembrane transport of substances and migration of cells. Therefore, it can be concluded that membrane proteins are very highlighted molecular changes in both the formation and evolution of GBC from another point of view.

To elucidate the molecular mechanism of GBC carcinogenesis and progression, we proposed the following hypotheses. The metabolic changes mainly related to lipid induce gallbladder stones, which stimulate the gallbladder wall for a long time, cause damage to the gallbladder mucosa, and lead to chronic inflammation of the gallbladder wall. Chronic inflammation further induces the transformation of genes related to immune activity and connection between cells, leading to malignant proliferation of the gallbladder mucosa and evasion of the body's immune surveillance. Further changes in membrane proteins mainly related to substance transportation, lead to changes in the internal and external environment of cells and changes in the nature of cell migration, which promotes cancer cells to spread far away, especially through lymphatic metastasis. Inflammation plays a key role in these processes, and changes in membrane proteins are the most distinct molecular changes, as shown in Fig 8.

Our research highlights the roles of metabolism, inflammation, immunity, and membrane proteins in GBC development. However, it only provides an overview landscape; the specific detailed molecular mechanisms still require further study, which will help the development of targeted drugs and improve the prognosis of GBC.

## Supporting information

**S1 Fig. Comparison of sequencing data with the human genome.** (A) Compared with the human genome, the ratio of reads aligned to gene regions, coding regions, splice sites, introns

and non-coding regions was normal. (B) Saturation analysis indicated that the amount of sequencing was sufficient. (C) The sequencing results of samples N10, N20, N8, Y12, Y8, Y13 well covered the genome. (D) The sequencing results of samples T1 T19 T22 T27 T32 well covered the genome. (E) The sequencing results of samples Y16 T5 T12 T13 T18 T31 well covered the genome.
(PDF)

**S2 Fig. Comparison of the characteristics of mRNAs and lncRNAs.** (A) Comparison of the number of exons between lncRNAs and mRNAs. (B) Comparison of the length distribution of lncRNAs and mRNAs. (C) Comparison of the expression levels of lncRNAs and mRNAs: take the average of the expression values of each transcript of lncRNA and mRNA, and draw the box plot with the log10 (FPKM+1) values.
(PDF)

**S3 Fig. GO and KEGG enrichment analysis of differentially expressed target genes of differentially expressed lncRNAs between normal gallbladder and gallbladder with chronic inflammation.** (A) The top 30 GO terms with a high degree of enrichment, the shapes of icons represent different GO categories, the size represents the number of differentially expressed target genes of differentially expressed lncRNAs contained by this GO term, the color depth represents the size of the q-value, the X axis indicates the value of rich factor. (B) The 28 GO terms with q-value $\leq 0.05$ were further classified, numbers on the graph represent the number of GO terms corresponding to the category. (C) The top 30 KEGG terms with a high degree of enrichment. (D) The 16 KEGG terms with p-value $\leq 0.05$ were further classified, numbers on the graph represent the number of KEGG items corresponding to the category.
(PDF)

**S4 Fig. GO and KEGG enrichment analysis of target genes of differentially expressed lncRNAs between gallbladder with chronic inflammation and early gallbladder cancer.** (A) The top 30 GO terms with a high degree of enrichment, the shapes of icons represent different GO categories, the size represents the number of target genes of differentially expressed lncRNAs, the color depth represents the size of the q-value, and the X axis indicates the value of rich factor. (B) The 76 GO terms with p-value $\leq 0.05$ were further classified, numbers on the graph represent the number of GO terms corresponding to the category. (C)The top 30 KEGG terms with a high degree of enrichment. (D) The 17 KEGG terms with p-value $\leq 0.05$ were further classified, numbers on the graph represent the number of KEGG terms corresponding to the category.
(PDF)

**S5 Fig. GO and KEGG enrichment analysis of target genes of differentially expressed lncRNAs between early gallbladder cancer and advanced gallbladder cancer.** (A) The top 30 GO terms with a high degree of enrichment, the shapes of icons represent different GO categories, the size represents the number of target genes of differentially expressed lncRNAs, the color depth represents the size of the q-value, and the X axis indicates the value of rich factor. (B) The 50 GO terms with p-value $\leq 0.05$ were further classified, numbers on the graph represent the number of GO terms corresponding to the category. (C) The top 30 KEGG terms with a high degree of enrichment.
(PDF)

**S1 Table. Clinicopathological data of normal gallbladder and gallbladder with chronic inflammation.**
(DOCX)

**S2 Table. Clinicopathological data of gallbladder cancer.**
(DOCX)

**S3 Table. Reagents for library construction and quality inspection.**
(DOCX)

**S4 Table. Quality inspection results of library.**
(DOCX)

**S5 Table. Inspection results of sequencing data.**
(DOCX)

**S6 Table. Preprocessed statistics of sequencing.**
(DOCX)

**S7 Table. Genome mapping results.**
(DOCX)

**S8 Table. Primer sequences in the quantitative real-time PCR experiment.**
(DOCX)

## Acknowledgments

We thank Shengxian Yuan (Eastern Hepatobiliary Surgery Hospital, Shanghai, China) for giving us some useful advises on the study. We also thank Editage (www.editage.cn) for English language editing.

## Author Contributions

**Conceptualization:** Sen Yang.

**Data curation:** Sen Yang, Litao Qin, Yanling Zhang.

**Formal analysis:** Litao Qin, Bing Mao, Xueliang Yue.

**Funding acquisition:** Hongshan Liu.

**Investigation:** Sen Yang, Litao Qin, Pan Wu, Yanbing Liu, Yanling Zhang, Yiyang Yan, Shuai Yan, Feilong Tan.

**Methodology:** Litao Qin.

**Project administration:** Pan Wu.

**Resources:** Pan Wu, Yanbing Liu, Yiyang Yan, Shuai Yan, Feilong Tan, Xueliang Yue.

**Supervision:** Sen Yang, Huanzhou Xue.

**Validation:** Sen Yang.

**Visualization:** Bing Mao.

**Writing – original draft:** Sen Yang.

**Writing – review & editing:** Hongshan Liu, Huanzhou Xue.

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
