## [Decision Letter · Decision Letter 0]

7 Dec 2022

PONE-D-22-27414RNA sequencing revealed the multi-stage transcriptome transformations during the development of gallbladder cancer associated with chronic inflammationPLOS ONE

Dear Dr. Xue,

Thank you for submitting your manuscript to PLOS ONE. After careful consideration, we feel that it has merit but does not fully meet PLOS ONE’s publication criteria as it currently stands. Therefore, we invite you to submit a revised version of the manuscript that addresses the points raised during the review process.

This manuscript presents some interesting data but needs improvements in writing and presentation. Also, all comments of the reviewer need to be addressed and explained in author's response letter. In addition, authors need to indicate that all the data is available for sharing.

We look forward to receiving your revised manuscript.

Kind regards,

Ajay Pratap Singh, Ph.D.

Academic Editor

PLOS ONE

Journal Requirements:

3. You indicated that you had ethical approval for your study. Please clarify whether minors (participants under the age of 18 years) were included in this study. If yes, in your Methods section, please ensure you have also stated whether you obtained consent from parents or guardians of the minors included in the study or whether the research ethics committee or IRB specifically waived the need for their consent.

4. Please describe in your methods section how capacity to provide consent was determined for the participants in this study. Please also state whether your ethics committee or IRB approved this consent procedure. If you did not assess capacity to consent please briefly outline why this was not necessary in this case.

Reviewers' comments:

Reviewer's Responses to Questions

**Comments to the Author**

1. Is the manuscript technically sound, and do the data support the conclusions?

Reviewer #1: Yes

2. Has the statistical analysis been performed appropriately and rigorously? 

Reviewer #1: Yes

3. Have the authors made all data underlying the findings in their manuscript fully available?

Reviewer #1: Yes

4. Is the manuscript presented in an intelligible fashion and written in standard English?

Reviewer #1: Yes

5. Review Comments to the Author

Reviewer #1: In this study, the authors demonstrated the changes in mRNAs and lncRNAs expression during the evolution of GBC using next-generation RNA sequencing. This article includes interesting data, however, there are a couple of issues that need to be addressed.

1. In the case selection and sample processing description under the method section, the authors need to explain how they determined the number of patient cases for each group (normal, chronic inflammation, early stage and advanced stage GBC)

2. Figure 1 should be explained better in the text (result section). In addition, the rationale for selecting 12 genes for the qPCR should be stated. The ** in Figure 1 should be placed close to each other if they are depicting p ≤ 0.01 in order not to confuse the reader.

3. Figures 3, 5 and 7 are not well described in the text. Authors should explain each of the figures in the text in a comprehensive manner. Authors should be clear about what they mean by by GO/KEGG items. (If items means genes/pathways, it should be written clearly)

4. Figures 2B, 4B and 6B- The legends for these figures are not written correctly. Green dots needs to be changed to blue dots.

5. Figures 3A, S3A, S4A, 5A, S5A and 7A- It was mentioned in the legends for these figures that the shape represent different GO categories, however, there are other categories with similar shapes as well. The authors should look into this and make necessary corrections.

6. In the GO enrichment of differentially expressed mRNAs between normal gallbladder and gallbladders with chronic inflammation result, it was mentioned that the number of mRNAs with larger enrichment factor in the metabolism category is 24, however, only a total of 14 was shown for the metabolism category in the figure (Figure 3B). Thus, the authors need to ensure the description in the text matches the figures represented.

7. The results showed that transcriptomic differences between the inflammatory gallbladder and the normal gallbladder are related to inflammation and metabolism. Furthermore, changes related to immune activity were important in early GBC, which suggest that inflammation and immune activity play a role in the formation of GBC. However, in the discussion section, the authors stated that membrane proteins are the most highlighted molecular changes in the formation and evolution of GBC. Therefore, the authors need to provide a clear rationale why membrane proteins are the most highlighted molecular changes in the formation of GBC.

8. The second statement in the introduction section needs to be revised.

6. PLOS authors have the option to publish the peer review history of their article (what does this mean?). If published, this will include your full peer review and any attached files.

Reviewer #1: No

---

## [Author Response · Author response to Decision Letter 0]

11 Feb 2023

January 19, 2023

Dear Dr. Ajay Pratap Singh

Thank you for your recent review of our manuscript, “RNA sequencing revealed the multi-stage transcriptome transformations during the development of gallbladder cancer associated with chronic inflammation”（No. PONE-D-22-27414）. We have carefully considered each of the comments and have performed additional studies and analyses to address these comments. A point-by-point response follows.

Part 1: Journal Requirements

Requirements #1: Please ensure that your manuscript meets PLOS ONE's style requirements, including those for file naming. The PLOS ONE style templates can be found at https://journals.plos.org/plosone/s/file?id=wjVg/PLOSOne_ formatting_ sample _main_body.pdf and https://journals.plos.org/plosone/s/file?id=ba62/ PLOSOne_format ting_sample_title_authors_affiliations.pdf

Response: We have checked the style of our manuscript again and ensured the manuscript met PLOS ONE’s style.

Requirements #2: Please provide additional details regarding participant consent. In the ethics statement in the Methods and online submission information, please ensure that you have specified what type you obtained (for instance, written or verbal, and if verbal, how it was documented and witnessed). If your study included minors, state whether you obtained consent from parents or guardians. If the need for consent was waived by the ethics committee, please include this information.

Response: Written informed consents were obtained from all participants in this study. Minor was not included in our study and the age of participants was between 40 and 82 years (Revised Manuscript with Track Changes, line 111-115). 

Requirements #3: You indicated that you had ethical approval for your study. Please clarify whether minors (participants under the age of 18 years) were included in this study. If yes, in your Methods section, please ensure you have also stated whether you obtained consent from parents or guardians of the minors included in the study or whether the research ethics committee or IRB specifically waived the need for their consent.

Response: Our study was approved by the Ethics Committee of Henan Provincial People’s Hospital, and minors were not included in this study (Revised Manuscript with Track Changes, line 112-113). 

Requirements #4: Please describe in your methods section how capacity to provide consent was determined for the participants in this study. Please also state whether your ethics committee or IRB approved this consent procedure. If you did not assess capacity to consent please briefly outline why this was not necessary in this case.

Response: The capacity to provide consent of the participants was assessed by our researchers. All the participants were adults with age range between 40 and 82. They were in well performance status, spoke and understood Chinese, and were able to give informed consent. This consent procedure was approved by the Ethics Committee of Henan Provincial People’s Hospital. We improved the description in the methods section (Revised Manuscript with Track Changes, line 111-115). 

Requirements #5: Please review your reference list to ensure that it is complete and correct. If you have cited papers that have been retracted, please include the rationale for doing so in the manuscript text, or remove these references and replace them with relevant current references. Any changes to the reference list should be mentioned in the rebuttal letter that accompanies your revised manuscript. If you need to cite a retracted article, indicate the article’s retracted status in the References list and also include a citation and full reference for the retraction notice. 

Response: We reviewed our reference list again, and ensured that it was complete and correct. We increased 3 new references in the first paragraph of introduction (References 2，4，and 5).

Part 2: Reviewers' comments

We thank you for reviewer’s kindly work and great advice. We have carefully considered each of the reviewers’ comments and have performed additional studies and analyses to address these comments. A point-by-point response follows.

Comments #1: In the case selection and sample processing description under the method section, the authors need to explain how they determined the number of patient cases for each group (normal, chronic inflammation, early stage and advanced stage GBC).

Response: We thank you for reviewer’s kindly work. Repeatability is an important factor of scientific experiments, and at least three repetitions are required in scientific studies. In order to determine an appropriate number of samples, we considered the repeatability of experiments and the difficulty of obtaining samples. The normal gallbladder samples were the most difficult to obtain in our study. In addition, RNA sequencing experiment requires very high quality of samples, some samples with poor quality were excluded from the experiment. Considering the above factors, we determined the sample size for RNA sequencing, including 3 normal gallbladder, 4 inflammatory gallbladder, 5 early gallbladder cancer, and 5 advanced gallbladder cancer. This can basically meet the requirements of scientific experiments on repeatability. We have made corresponding modifications in the case selection and sample processing description under the method section (Revised Manuscript with Track Changes, line 119-122).

Comments #2: Figure 1 should be explained better in the text (result section). In addition, the rationale for selecting 12 genes for the qPCR should be stated. The ** in Figure 1 should be placed close to each other if they are depicting p ≤ 0.01 in order not to confuse the reader.

Response: Thank you for reviewer’s great advice. We have modified the first part of the results and Figure 1. The 12 genes with significant expression differences between different groups were selected to performed qPCR to verify the results of RNA-seq. Briefly, we selected two mRNAs and two lncRNAs for each comparison. This was equivalent to that we verified 4 genes for early gallbladder cancer, 8 genes for gallbladder with chronic inflammation and early gallbladder cancer, and 4 genes for advanced gallbladder cancer. In total, 104 sequencing data have been verified. (Revised Manuscript with Track Changes, line 276-284).

Comments #3: Figures 3, 5 and 7 are not well described in the text. Authors should explain each of the figures in the text in a comprehensive manner. Authors should be clear about what they mean by by GO/KEGG items. (If items means genes/pathways, it should be written clearly).

Response: Thank you for reviewer’s insightful suggestion. We have improved the description of Figure 3, 5 and 7 in the text. We thought the word "GO item" and “KEGG item” were not accurate, so we decided to replace them with "GO term" and “KEGG term” which were also adopted by GO and KEGG websites (Revised Manuscript with Track Changes, line 320-325, 347-355，395-400，421-426，466-476，and 493-497). 

In the text GO term refers to the smallest set of genes with similar functions in GO database. It is not a single gene, but the smallest category of gene classification in GO. For example, estrogen 16- α- Hydroxylase activity is a GO term, which refers to a collection of genes participating in estrogen 16- α- Hydroxylase activity. KEGG term refers to the minimum category of pathways in KEGG database. For example, synthesis and degradation of ketone body is a KEGG term which refers to all the pathways related to synthesis and degradation of ketone body. The above are also illustrated in the GO and KEGG websites.

Comments #4: Figures 2B, 4B and 6B- The legends for these figures are not written correctly. Green dots needs to be changed to blue dots.

Response: Thank you for reviewer’s carefully work. We have made corresponding modifications in these legends (Revised Manuscript with Track Changes, line 314，389，and 459).

Comments #5: Figures 3A, S3A, S4A, 5A, S5A and 7A- It was mentioned in the legends for these figures that the shape represent different GO categories, however, there are other categories with similar shapes as well. The authors should look into this and make necessary corrections.

Response: Thank you for reviewer’s kindly work. GO categories include biological process, cellular component, and molecular function. In the figures of previous manuscript, we used three different shapes to represent them. But we inadvertently used inconsistent icons in these figures. Therefore, we modified the icons of figures 3A 5A to keep consistent with figures 7A S3A S4A S5A in the revised manuscript (Revised Manuscript with Track Changes, figures 3A and 5A).

Comments #6: In the GO enrichment of differentially expressed mRNAs between normal gallbladder and gallbladders with chronic inflammation result, it was mentioned that the number of mRNAs with larger enrichment factor in the metabolism category is 24, however, only a total of 14 was shown for the metabolism category in the figure (Figure 3B). Thus, the authors need to ensure the description in the text matches the figures represented.

Response: Thank you for reviewer’s carefully work. The correct number is 14. We have made corresponding modification in the text (Revised Manuscript with Track Changes, line 326).

Comments #7: The results showed that transcriptomic differences between the inflammatory gallbladder and the normal gallbladder are related to inflammation and metabolism. Furthermore, changes related to immune activity were important in early GBC, which suggest that inflammation and immune activity play a role in the formation of GBC. However, in the discussion section, the authors stated that membrane proteins are the most highlighted molecular changes in the formation and evolution of GBC. Therefore, the authors need to provide a clear rationale why membrane proteins are the most highlighted molecular changes in the formation of GBC.

Response: Thank you for reviewer’s insightful suggestion. Actually, we explained this point in the previous manuscript, line 491-495. However, the expression was not so clear and rigorous. Therefore, we improved the expression in the revised manuscript. Briefly, membrane proteins play extremely role in the activity of inflammation, immunity, connection between cells, transmembrane transport of substances and migration of cells, which are important changes in the formation and evolution of GBC. Therefore, we concluded that membrane proteins were very highlighted molecular changes in both the formation and evolution of GBC from another point of view. (Revised Manuscript with Track Changes, line 561-570, and 64).

Comments #8: The second statement in the introduction section needs to be revised.

Response: Thank you for reviewer’s great advice. We cited 3 new references to make our statement more convincing (References 2，4，and 5)，and improved and modified the description in the introduction (Revised Manuscript with Track Changes, line 68-75).

---

## [Editor Report · Decision Letter 1]

17 Mar 2023

RNA sequencing revealed the multi-stage transcriptome transformations during the development of gallbladder cancer associated with chronic inflammation

PONE-D-22-27414R1

Dear Dr. Xue,

We’re pleased to inform you that your manuscript has been judged scientifically suitable for publication and will be formally accepted for publication once it meets all outstanding technical requirements.

Kind regards,

Ajay Pratap Singh, Ph.D.

Academic Editor

PLOS ONE
---

## [Editor Report · Acceptance letter]

22 Mar 2023

PONE-D-22-27414R1 

RNA sequencing revealed the multi-stage transcriptome transformations during the development of gallbladder cancer associated with chronic inflammation 

Dear Dr. Xue:

I'm pleased to inform you that your manuscript has been deemed suitable for publication in PLOS ONE. Congratulations! Your manuscript is now with our production department. 

Kind regards, 

on behalf of

Dr. Ajay Pratap Singh 

Academic Editor

PLOS ONE